# Divergent Last Century Tree Growth along an Altitudinal Gradient in A *Pinus sylvestris* Dry-edge Population

**Laura Fernández-Pérez [1],\*, Miguel Ángel Zavala [1], Pedro Villar-Salvador [1] and Jaime Madrigal-González [2]**

[1] Forest Ecology and Restoration Group, Universidad de Alcalá, Alcalá de Henares, 28801 Madrid, Spain; ma.zavala@uah.es (M.Z.); pedro.villar@uah.es (P.V.-S.)

[2] Climate Change Impacts and Risks in the Anthropocene, Institute for Environmental Sciences, University of Geneva, 66 Boulevard Carl Vogt, 1205-CH Geneva, Switzerland, ecojmg@hotmail.es

\* Correspondence: laura.fernandezp@edu.uah.es

**Abstract:** Research Highlights: This research highlights the importance of environmental gradients in shaping tree growth responses to global change drivers and the difficulty of attributing impacts to a single directional driver. Background and Objectives: Temperature increases associated with climate change might strongly influence tree growth and forest productivity in temperate forest species. However, the direction and intensity of these effects at the dry edge of species range are still unclear, particularly given the interaction between local factors and other global change drivers such as land use change, atmospheric $CO_2$ increase and nitrogen deposition. While recent studies suggest that tree growth in cool temperate forests has accelerated during the last decades of the 20th century, other studies suggest a prevalence of declining growth, especially in dry-edge populations. Materials and Methods: Using historical forest inventories, we analyzed last century tree growth trends (1930–2010) along an elevation gradient (1350–1900 meters above sea level, (m a.s.l.)) in a dry edge scots pine (*Pinus sylvestris* L.) forest in Central Iberian Peninsula. Growth was estimated as decadal volume increments in harvested trees of different size classes from 1930 to 2010 (1930–1940, 1939–1949, 1949–1959, 1959–1968, 1989–1999, 2000–2010). *Results:* Our results showed opposite growth trends over time depending on elevation. While tree growth has accelerated in the low end of the altitudinal gradient, tree growth slowed down at higher elevations (1624–1895 m a.s.l.). Moreover, the magnitude of growth reduction along the altitudinal gradient increased with tree age. Conclusions: Throughout the last 80 years, growth trends in a rear-edge *Pinus. sylvestris* forest has shown divergent patterns along an altitudinal gradient. Specifically, environmental conditions have become more adverse for growth at high altitudes and have improved at low altitudes. This suggests that local factors such as topography can modulate the impact of climate change on forest ecosystems.

**Keywords:** altitude; global change; high mountain; Mediterranean; Scots pine

## 1. Introduction

Global change drivers such as the increase in atmospheric $CO_2$ concentrations ($C_a$), increases mean temperature or nitrogen (N) deposition might affect forest productivity in temperate forests [1–3]. A higher $C_a$ and soil N may enhance net photosynthesis but have variable effects on stomatal conductance. If photosynthesis increases more rapidly than stomatal conductance, it can increase plant water use efficiency [4,5]. Experiments using $CO_2$ enriched environments support this

hypothesis and show that the increment of photosynthesis and water use efficiency in tree species in response to higher $C_a$ is mainly due to increased carboxylation [6]. Tree ring studies, models and inventory data indicate that tree growth has increased in boreal and temperate forest ecosystems during the last decades providing indirect support to the $C_a$ and N "fertilizing" effects [7,8]. In conjunction with a rise in $C_a$, aridity has increased during the last decades in many regions [9,10]. It is therefore unclear whether "fertilization" effects can compensate or exceed the costs of increased aridity in drought-prone ecosystems (e.g. [11,12]). Moreover, this balance can also be critically modulated by other local factors such as competition, evapotranspiration rate, or forest management (see for example [13,14]).

In water-limited Mediterranean ecosystems, previous findings have not shown clear tree growth trends with respect to time and associated drivers. While some studies have reported a neutral [15] or even a negative relation between tree growth and rising $C_a$ [16], other studies have reported positive effects [14,17]. Differences among studies could be due to local environmental conditions- which offset the effects of $C_a$ and N deposition on tree growth- and/or to species individual functional responses, which may show different responsiveness along resource and environmental gradients [18]. Mountain Mediterranean forests represent natural ecotones dominated by Eurosiberian tree species at the edge between the semi-arid and cool temperate biomes. That is why mountain forests in the Mediterranean are considered highly vulnerable to global change, including tree species such as *Pinus sylvestris* L. (Scots pine), which is in the southern limit of its range [19]. During the last decades, the intensity of droughts has increased, reducing tree productivity in drought-prone environments [20]. In the Mediterranean basin, the future climatic scenarios predict a temperature increase and a reduction of precipitation [21]. Tree populations living at the dry edge of species distribution are often vulnerable to rapid climate changes, and particularly to intense drought spells [22]. Accordingly, recent studies conducted in dry-edge populations provide evidence of a reduction in tree growth and in other demographic parameters possibly caused by increasing aridity (e.g. [23]). Further evidence, however, is needed to understand how global change impacts these ecosystems and, in particular, how these impacts are modulated by both endogenous (such as tree age) and exogenous (such as topography) factors. This knowledge is critical in order to unveil the mechanisms underlying forest resistance and resilience to the impact of global change and to develop efficient adaptation measures for population persistence.

*Pinus sylvestris* is a widely distributed in the northern hemisphere having its dry edge of distribution in mountain areas of southern Europe [24]. Several studies have focused on the abiotic determinants of *Pinus. sylvestris* tree growth across its distribution range [25,26]. Particularly, drought is a key factor driving growth in *Pinus. sylvestris* in Mediterranean-climate mountains [11], albeit low winter temperatures can also play an important role [27]. Global change drivers (i.e., warming winters and $C_a$ and N fertilization effects) might have increased *Pinus. sylvestris* growth throughout the last century, yet this hypothesis has little support in Mediterranean mountains [11].

In this study, we analyzed last century tree growth trends in an extensive Mediterranean *Pinus. sylvestris* forest located in the Central System range in the Iberian Peninsula where the species is at the dry edge of its distribution. Specifically, we examined whether tree growth has accelerated over time as expected from increased $C_a$, and N "fertilization" effects and milder winters, or, on the contrary, whether tree growth has decreased due to increased aridity. *Pinus sylvestris* forests in the Central System range are under a typical mountain Mediterranean climate within an altitudinal gradient that ranges from 1300 to 2100 m a.s.l. We examined tree growth responses along an altitudinal gradient as a surrogate of environmental variation and throughout the period 1930–2010 which encompasses several global change drivers correlated with time.

## 2. Materials and Methods

### 2.1. Study Area

Tree growth was recorded in a 3891 ha forest (named MUP 198 "Pinar de Navafría") dominated by *Pinus. sylvestris* and located in the municipality of Navafría (Lat 41°03'17'' N; Long 3°49'21'' W) in

the Sierra de Guadarrama, Central System Range, Spain (Figure 1). In this area, *Pinus. sylvestris* is bound to a narrow altitudinal strip ranging from 1300 to 2100 m a.s.l. (Table A1, [28]). Climate is continental Mediterranean with important altitudinal variations in temperature and rainfall. Winters are cold and humid, and the summers are warm and dry. At the lowest part of the altitudinal gradient (the village of Navafría, 1192 m a.s.l.), the annual, minimum and maximum mean temperatures are 10.7, 4.4 and 17 °C, respectively. Total annual rainfall average is 611 mm and no apparent temporal trend is observed during the study period (Figure A1). Similar to other mountains, temperature decreases with elevation and rainfall follows a spatially more variable pattern [29]. In Puerto of Navacerrada (1894 m a.s.l.), total annual rainfall is 1324 mm and the mean annual temperature is 6.4 °C. In Puerto of Navacerrada, a slight decrease in total annual rainfall is recognized while in Segovia city (~1000 m a.s.l.), the precipitation has remained constant throughout the years (Table A1). In contrast, the annual temperature has increased in the last 50 years in both locations.

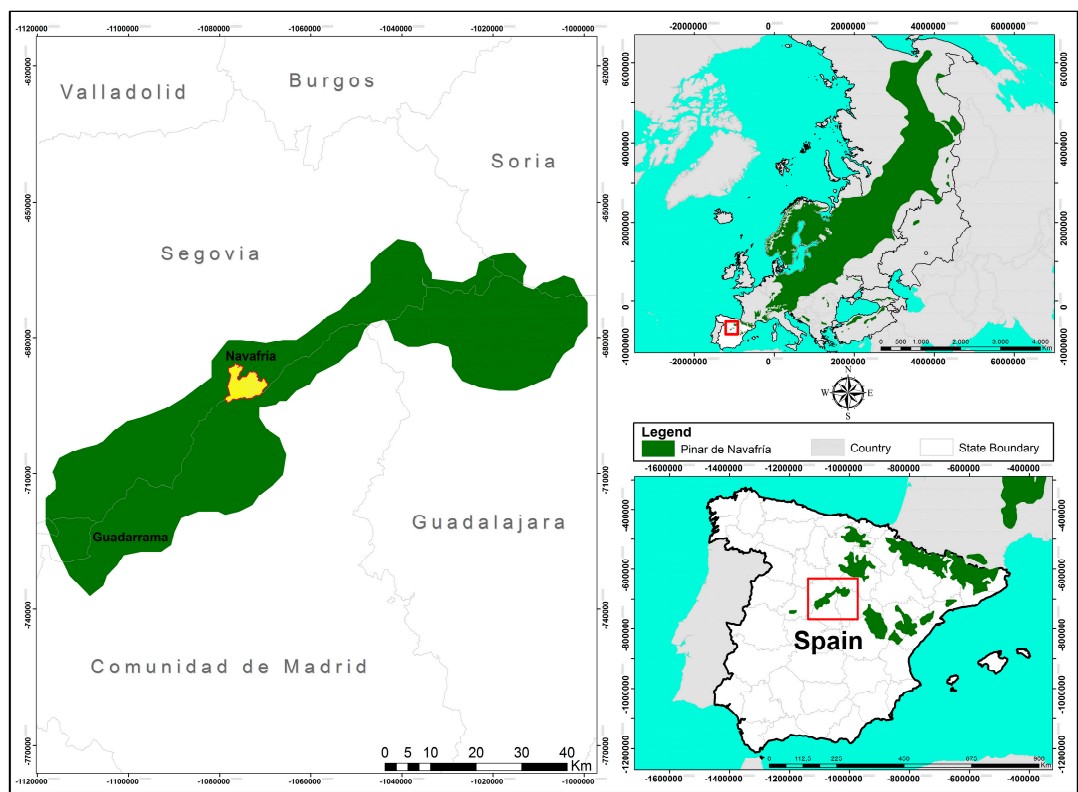

**Figure 1.** Distribution of *Pinus sylvestris* in Europe and Spain (**right**) and location of the forest Pinar de Navafría in the Central System range (**left**).

### 2.2. Tree Size and Growth Measurements

Scots pine forest management in Sierra de Guadarrama has been subjected to management plans since 1890, mainly oriented towards sustainable timber production. The "Pinar de Navafría" forest began to be managed in 1885 using a shelterwood thinning methods to maintain natural regeneration and the production of high-quality timber [30]. For administrative purposes, the forest was divided into nine sections with identical management plans. Each section in turn was subdivided into five subsections, thus resulting in a total of 45 forest units. Rotation averages 120 years and the regeneration period spans 20 years. Periodic thinning is conducted every 10 years depending on the stage of the rotation period and the stand is managed in order to achieve even-aged stands for the long term.

Forest management prescriptions have been audited almost every ten years, when trees of different diameter classes were selected and harvested for estimating dendrometric tree variables. Specifically, these variables were tree age and stem volume, which have been recorded since 1890, thus providing an invaluable source of information to assess the evolution of tree structure and growth throughout the last century. All data were recorded in a regional historical archive. Tree growth data in particular are available for decadal periods in 1930–1940, 1939–1949, 1949–1959, 1959–1968, 1989–1999, 2000–2010. In the first record (1930–1940), the present and definitive stands were defined, preventing the use of the data from the first management plan (1885–1895). Slope aspect, area, altitude and soil type were recorded for each forest unit. For each fallen tree the diameter at breast height after removing the bark (dbh, cm), height (m) and age (number of rings) was measured (Table 1). Then the trunk was transversally cut into small cylindrical segments to measure the volume without bark assuming the segments to be cylinders. The total volume of the trunk (V) was calculated as the sum of the volume of each trunk segment. The volume of each tree 10 years before (V') was calculated in a similar way by subtracting the growth of the last 10 tree-ring width. Absolute growth was estimated as the difference between trunk volumes (V-V'). Annual growth rate was estimated for each tree as the absolute growth divided by the number of years. The age of the studied trees ranged between 30 and 235 years old.

**Table 1.** Characteristics of *Pinus. sylvestris* in the study area. Density values are number of individuals per ha. Values in parenthesis are standard deviation.

| Year | No. trees | DBH | Average height | Average age | Min age | Max age | Density (no. ind/ha) |
|------|-----------|-----|----------------|-------------|---------|---------|----------------------|
| 1930 | 109 | 39.53 (13.57) | 14.08 | 92 | 32 | 156 | 178.26 (64.7) |
| 1940 | 132 | 39.65 (13.32) | 13.68 | 95 | 31 | 235 | 258.6 (101.34) |
| 1950 | 96 | 37.28 (14.43) | 12.82 | 89 | 40 | 172 | 302.21 (128.1) |
| 1959 | 73 | 42.61 (12.86) | 12.53 | 101 | 40 | 235 | 398.41 (140.87) |
| 1990 | 83 | 40.35 (12.65) | 12.38 | 96 | 39 | 181 | 405.55 (86.7) |
| 2000 | 116 | 36.17 (11.66) | 14.44 | 94 | 30 | 235 | 369.08 (89.67) |
| **Total** | **609** | | | | | | |

*2.3. Data Analysis*

We used the platform R to perform the statistical analyses [31]. We used Linear Mixed Models (LMM) of the nlme package [32] to analyze tree growth as a function of the three-order interaction altitude × calendar years × tree age to evaluate whether global-change trends of tree growth are dependent on tree age and altitude since 1930. To avoid collinearity between the variables, we firstly analyzed correlation among variables using Pearson correlation with the Corrplot function in the Hmisc package. Selected predictor variables were all uncorrelated and thus multicollinearity is expected to be negligible in the LMM (Table 2).

**Table 2.** Correlation among selected predictor variables.

| Variables | Year | Altitude | Stand Density | Trunk diameter | Tree Age |
|-----------|------|----------|---------------|----------------|----------|
| Altitude | -0.04 | 1 | - | - | - |
| Stand density | 0.39 | -0.37 | 1 | - | - |
| Trunk diameter | -0.06 | -0.01 | -0.04 | 1 | - |
| Tree age | -0.02 | 0.07 | -0.04 | 0.70 | 1 |

Importantly, tree density depicted a somewhat linear increment throughout the studied period (Figure 2) so we decided to fit a first model with only this variable to remove its effects prior to analyzing long-term linear trends using the calendar years. To do that, we used the lme4 statistical package in the R environment [32]. The model was developed as follows:

$$Y_{ijt} = \beta_0 + \beta_1 \text{ Density } + b_1 + \varepsilon_{ijt} \tag{1}$$

where Y represents annual growth (untransformed), *i* is the power of the linear dimension associated to the variable, *j* is the identifier of the variable and *t* is age. The fixed parameters are $\beta_0 - \beta_4$, while $b_1$ is random effect at forest unit level ($b_1 \sim N(0, \tau_1^2)$), and finally $\varepsilon_{ijt}$ shows the error $\varepsilon_{ijt} \sim N(0, \tau_1^2)$.

We thereafter developed the backward selection models using the residuals of the previous model as the dependent variable and the third order interaction Altitude × Calendar year × tree age as the fixed effects term (Table 3). The biological meaning of this third-order interaction is that depending on the tree age, and altitude, differential global change trends associated with rising $C_a$, N or temperature should be recognized in tree growth. We used Maximum Likelihood method for parameter estimation in the fixed effects selection [33]. All potential candidate models were compared by using Akaike Information Criterion (AICc). Following a general rule of parsimony, models exceeding 2 units of the minimum AICc were not considered for inference. If the set of contemporary trees develops differently in different years, it will indicate changes in growth and site conditions. The model with the most appropriate structure was as follows:

$$Z_{ijt} = \beta_0 + \beta_1 \, \text{Age} + \beta_2 \, \text{Altitude} + \beta_3 \, \text{Year} + \beta_4 \, \text{Age} \times \text{Year} + \beta_5 \, \text{Altitude} \times \text{Year} + b_1 + \varepsilon_{ijt} \tag{2}$$

$Z_{ijt}$ represents residuals of model 1. The fixed parameters are $\beta_0 - \beta_5$, while $b_1$ is random effect ($b_1 \sim N(0, \tau_1^2)$), and finally $\varepsilon_{ijt}$ shows the error $\varepsilon_{ijt} \sim N(0, \tau_1^2)$. We considered forest units as a random factor that accounts for spatial dependencies associated to similar management status and soil conditions, among other potential unaccounted spatial contingencies that make trees within the same forest unit more similar than those from other units.

**Table 3.** Selection of the predictor variables using the corrected Akaike Information Criterion for small sample sizes (AICc).

| Step | Model | Fixed effects | Param | AICc | Δ AICC |
|------|-------|---------------|-------|------|--------|
| 1# | Y × Alt × Ag | Full model | 8 | 1238.042 | 0 |
| | Y × Alt + Ag × Y + Alt × Ag | Removed Y × Alt × Ag | 7 | 1232.405 | 5.637 |
| 2# | Y × Alt + Ag × Y + Alt × Ag | Removed Y × Alt × Ag | 7 | 1232.405 | 0 |
| | Y × Alt + Ag × Y | Removed Ag × Alt | 6 | 1234.160 | −1.75 |
| | Ag × Alt + Ag × Y | Removed Y × Alt | 6 | 1225.914 | 6.491 |
| | Ag × Alt + Y × Alt | Removed Ag × Y | 6 | 1234.757 | −2.352 |
| 3# | Null | intercept | 1 | 1700.557 | 0 |
| | Supported | Ag × Alt + Alt × Y | 6 | 1225.914 | 474.643 |

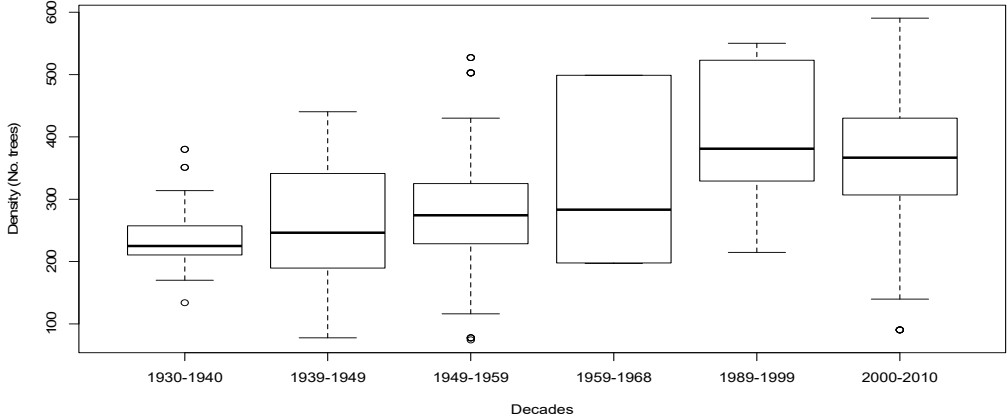

**Figure 2.** Tree density (in the 45 units) increase in the last decades in Navafría, Sierra the Guadarrama, Spain.

## 3. Results

The effect of the interactions Age × Altitude and Year × Altitude were both significant (Table 4). At low altitudes (1350–1623 m a.s.l.), tree growth increased over time, while at high altitudes (1624–1892 m a.s.l.) growth slowed down (Figure 3 and Table 4). Growth change across the altitudinal gradient was age–dependent; young trees showed overall low growth with no differences along the altitudinal gradient. As trees became older, they grew more slowly at high altitudes than at the low altitude sites, where growth was high (Figure 4). The correlation among the model parameters was negligible (Appendix 3).

**Table 4.** Parameters estimated for the double interaction model including associated standard error, t statistic and p value.

| Model parameters | Parameter value | Std.Error | t-value | P value |
|---|---|---|---|---|
| Intercept | -0.148 | 0.024 | -6.09 | <0.0001 |
| Age | 0.156 | 0.024 | 6.37 | <0.0001 |
| Altitude | -0.115 | 0.025 | -4.63 | <0.0001 |
| Year | -0.042 | 0.024 | -1.88 | 0.082 |
| Age × Altitude | -0.089 | 0.025 | -3.60 | 0.0004 |
| Altitude × Year | -0.063 | 0.025 | -2.49 | 0.012 |

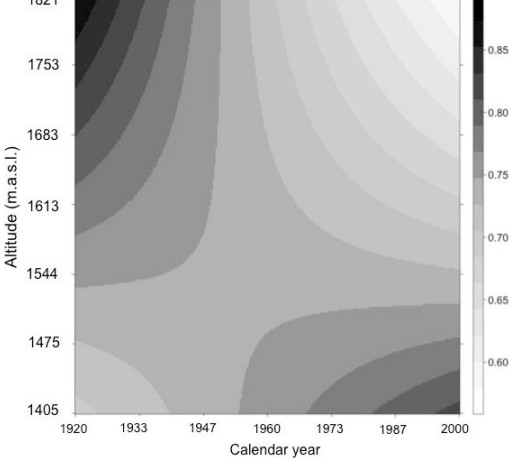

**Figure 3.** Contour plots showing tree growth responses (cm$^3$) to pair-wise interactions over the years and the altitude.

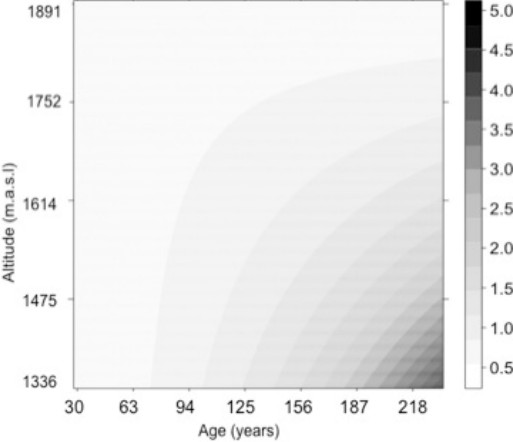

**Figure 4.** Contour plots showing tree growth responses (cm$^3$) to pair-wise interactions over the tree age and the altitude gradient.

## 4. Discussion

Our results suggest divergent last century *Pinus. sylvestris* growth patterns along the altitudinal gradient. While growth has increased notably over time at low altitudes, growth at high altitudes has significantly decreased. This result is somehow opposite to global expectations which emerged from trends reported in cool temperate and boreal forests, and which point to net positive effects of climate warming and $CO_2$ enrichment in forests in which low temperature is the predominant environmental constraint. Our results also showed that tree age modulated these responses along the altitudinal gradient. Specifically, growth divergence was more evident along the altitudinal gradient for older trees. Growth reduction with altitude is commonly seen in many mountains of the world [34]. Temperature decrease reduces plant metabolism and shortens the growing season [35,36], which reduces tree growth. A low growth rate and the lack of altitudinal gradient effects for young trees suggest that they experience similar stress level along the entire gradient. Growth at low altitudes might be constrained by competition and drought stress [37], while at high altitudes cold stress might be the main stressor for young trees.

Mean temperature has increased throughout the 20th century but especially in the last 20 years [38–40]. While some studies have reported growth acceleration at high elevation locations [41,42], others have shown no change or even a negative response [43,44]. In Sierra de Guadarrama, mean annual and winter temperatures have increased by 19% (5.75 °C to 6.87 °C) and 127% (–1.38 °C to 0.38 °C), respectively in the second half of the 20th century [45]. The increase in N deposition and atmospheric $C_a$ from the industrial era has also been proposed to enhance tree growth in some boreal conifers [8,46,47]. Field experiments of N enrichment point to modest increase in forest growth [46]. Specifically, for *Pinus. sylvestris* boreal stands, N enrichment increased growth by 12% [48]. In contrast, others studies shown no effect of N deposition [49]. In Sierra of Guadarrama, the N deposition showed exceedances due to its proximity to a large metropolitan area (~70 km from the metropolitan area of Madrid, [50]). It is possible that the temperature, N deposition and $C_a$ increase over time have stimulated tree growth at low altitudes due to enhanced water use efficiency [51]. Paradoxically, at high altitudes where low temperatures are the major limitation for plant performance [27], growth has decreased over time (Figure 3), suggesting lack of "fertilization" and warming temperature benefits on tree growth. This point outs that the benefits of these global change drivers have been overtaken out by unaccounted local factors operating at high altitudes. Consistent with our findings, a poor relation between tree growth and main global change drivers has been reported for other Mediterranean pine forests [11,14], providing further support to the idea that other unknown factors can mask the positive effects of warming and $C_a$ and N deposition "fertilization" effects.

Several local factors might explain the patterns described in our study. First, soils at high altitude may have lower water holding capacity and fertility than sites at the low end of the gradient [52,53]. Temperature increase over time involves higher evapotranspiration that, along with a slight reduction in precipitation at the high elevations (see Appendix B) may result in increasing drought stress [44,54]. A growth reduction has been reported in *Pinus. sylvestris* populations located at high altitudes, mainly associated with moisture conditions [55], suggesting that in Mediterranean mountains Scots Pine populations at high elevations may in fact be more vulnerable to drought stress than lower populations (however see [27]). Second, water availability rather than low temperature is the main limiting factor for tree growth at low- and mid-altitude sites in Mediterranean mountains [34,56]. *Pinus sylvestris* populations at low altitudes may be locally adapted and more drought tolerant than high altitude populations. Both altitudinal and latitudinal drought tolerance differences among *Pinus. sylvestris* populations have been reported [57,58]. Ecotypic differences in drought tolerance of tree species can be found even at short spatial scales (see [59]). Moreover, many plant species show ecotypic variations along altitudinal gradients with high altitude ecotypes growing slower than low altitude ecotypes due to inherent differences in metabolism, phenology and stress tolerance [60–62]. Thus, the negative effects of low temperatures with altitude can interact

synergistically with ecotype differences to reduce tree growth [58,63]. Finally, a third explanation is that *Pinus. sylvestris* populations at high altitudes are more sensitive to late spring frosts. Spring frosts might damage seedling emergence and hinder tree growth and survival [64]. Cold hardening and dehardening in many conifers are partially controlled by temperature [65], and mild winter spells can reduce cold hardening due to reduction in the needle concentration of the cryoprotectant soluble sugars [66,67]. It is possible that the increase in winter temperature may have advanced cold dehardening, making aboveground organs more vulnerable to spring frosts, whose frequency has remained stable as mean surface temperature has increased [68]. Increased damage caused by spring frosts due to premature dehardening has been observed for conifers [66,69] and for broadleaf trees [70].

## 5. Conclusions

Throughout the last 80 years a rear-edge *Pinus. sylvestris* forest has shown divergent growth patterns along an altitudinal gradient. Tree growth at low altitudes has increased in the second half of the 20th century, while at high altitudes it has slightly decreased. This suggests that environmental conditions and other factors may have become more adverse for growth at high altitudes and have improved at low altitudes. The positive effects of temperature, $C_a$ and N deposition increase on tree growth at low altitudes may have been greater than the potential negative effects of increased summer aridity. This study highlights the importance of local factors such as topography, which may interact with global scale environmental factors, to drive forest responses suggesting the need to deepen the monitoring of the factors associated with global change (e.g., microclimate, soil properties, disturbance regime, biotic pests, etc.) in order to understand similar impacts of climate change on these forests.

**Author Contributions:** conceptualization, Jaime Madrigal-González, Miguel A. Zavala, Laura Fernández-Pérez, Pedro Villar-Salvador; methodology, Jaime Madrigal-González and Laura Fernández-Pérez; formal analysis, Jaime Madrigal-González and Laura Fernández-Pérez; investigation, Jaime Madrigal-González and Laura Fernández-Pérez; resources, Miguel A. Zavala; data curation, Laura Fernández-Pérez; writing—original draft preparation, Laura Fernández-Pérez; writing—review and editing, Laura Fernández-Pérez, Jaime Madrigal-González, Miguel A. Zavala, Pedro Villar-Salvador; supervision Miguel A. Zavala, Pedro Villar-Salvador; project administration, Miguel A. Zavala; funding acquisition, Miguel A. Zavala".

**Funding:** This project was funded by grant FUNDIVER (MINECO, CGL2015-69186-311 C2-2-R). LFP acknowledges support from Consejo Nacional de Ciencia y Tecnología (CONACyT) of Mexico through scholarship 218637 for PhD studies. The authors acknowledge Raquel Lázaro Gutiérrez (Departamento de Filología Inglesa of Universidad de Alcalá) for the grammar revision.

**Acknowledgments:** We thank the Servicio Territorial de Medioambiente de Segovia (María Bragado Jambrina and José Ignacio Quintanilla Rubio; Segovia, Spain) for their help with data acquisition and their support with historical archives and Denis Conrado da Cruz for map elaboration. We thank Alistair Jump for his helpful comments on an earlier version of the manuscript.

**Conflicts of Interest:** The authors declare no conflict of interest. The funders had no role in the design of the study; in the collection, analyses, or interpretation of data; in the writing of the manuscript, or in the decision to publish the results.

## Appendix A.

**Table A1.** Altitude of the sections located in Navafría.

| . | | Elevation (m a.s.l.) | | |
|---|---|---|---|---|
| Section | Area (ha) | Min | Max | Mean |
| I | 1051 | 1320 | 2010 | 1620 |
| II | 904 | 1270 | 1990 | 1554 |
| III | 758 | 1330 | 2070 | 1700 |

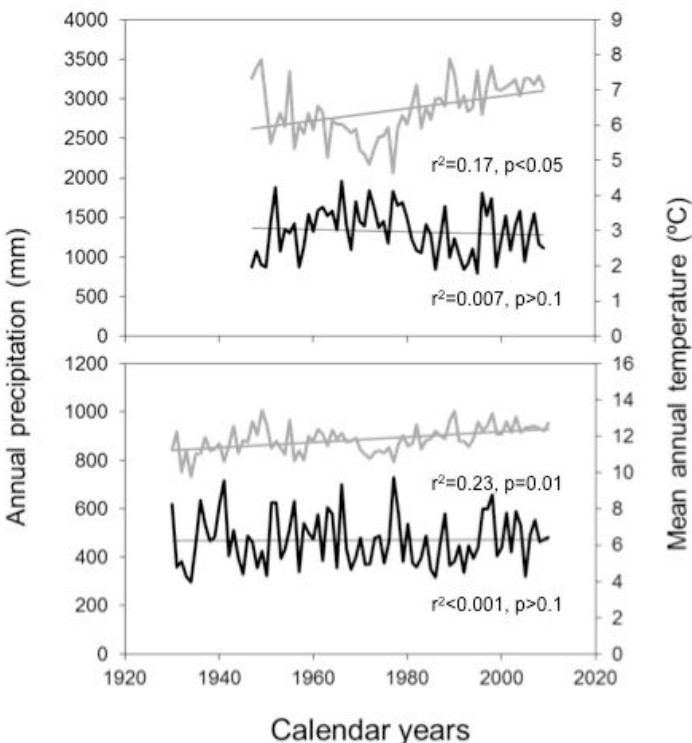

**Figure A1.** Annual rainfall (black lines) and mean temperature (grey line) through time in the Puerto de Navacerrada (upper panel) and Segovia city (bottom panel) located at 1894 m a.s.l. and 1002 m a.s.l. respectively (Data from AEMET).

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
