# Peer review of "Divergent Last Century Tree Growth along An Altitudinal Gradient in A Pinus sylvestris L. Dry-edge Population"

_forests, doi:10.3390/f10070532_

Reviewer 1 Report

Fernández-Pérez et al. present a very interesting manuscript on Scots pine at its most southern distribution. They show how growth performance changed with climate change. The manuscript is very well elaborated, the analysis is valid and the conclusion drawn is reasonable. The arguing throughout the manuscript is very easy to follow. I especially like, that the author kept the manuscript short and condensed the information to the necessary. It stands therefore as a nice report to that what is going on with Scots pine in higher altitudes in the Iberian Peninsula.

In summary, I recommend that some more detail, especially threats to the species under climate change should be given in the introduction or discussion (see minor comments). 

Minor comments:

Line 75 – 82: You could mention that scots pine is very important timber species, but it is in serious trouble elsewhere in its natural range due to extremely dry and hot summers and pathogen outbreaks and massive die-off (e.g. “Kunert, N. 2019. Das Ender der Kiefer in Mittelfranken als Hauptbaumart In Mittelfranken. AFZ – Der Wald 3, 24-25.“ Would be a very suitable reference for this, unfortunately only available in German). Hence, seeing that the species perforce better in some areas is very interesting to see.

Line 138: Rotation period of 120 years is very long for Scots pine, can you somehow make a rough estimate how rotation periods decreased in the low altitudes and increased in the high altitudes? Maybe mention this in the discussion?

Line 247: Yes, I think spring frost is an important factor, do have any numeric evidence for this (a rough guess will do)? How many more days is the on average frost in the high altitudes than in the low altitudes?

Author Response

We thank the Reviewer 1 for his positive comments. We have considered her/his minor comments as follow:

Line 75 – 82: You could mention that scots pine is very important timber species, but it is in serious trouble elsewhere in its natural range due to extremely dry and hot summers and pathogen outbreaks and massive die-off (e.g. “Kunert, N. 2019. Das Ender der Kiefer in Mittelfranken als Hauptbaumart In Mittelfranken. AFZ – Der Wald 3, 24-25.“ Would be a very suitable reference for this, unfortunately only available in German). Hence, seeing that the species perforce better in some areas is very interesting to see.

RESPONSE: We thank the reviewer for this interesting point. It would be really interesting to evaluate Scots pine health status across its range although. There is also evidence of decay ion some Iberian forests. These ideas are included in Reference 19 that describes Scots pine as a vulnerable species in Iberia. We have specified now that Scots pine in Iberia is at its southern range limit.

Line 138: Rotation period of 120 years is very long for Scots pine, can you somehow make a rough estimate how rotation periods decreased in the low altitudes and increased in the high altitudes? May be mention this in the discussion?

RESPONSE: This is an important issue as if there were strong differences in productivity or site quality along the altitudinal gradient, foresters should have adjusted the rotation period. According to historical records and current evidence this period however was maintained the same across the range. Tt is the regeneration period, however what can be very variable across years and sites.

Line 247: Yes, I think spring frost is an important factor, do have any numeric evidence for this (a rough guess will do)? How many more days is the on average frost in the high altitudes than in the low altitudes?

RESPONSE: This s a very interesting issue that can be also critical for co-dominance with other species such as hardwoods. Unfortunately we do not have exact meteorological data along the altitudinal gradient. We need to rely on using the altitudinal gradient as a surrogate for environmental variation. An interesting research avenue will be indeed describing variation in late frost events along the altitudinal gradient. This factor might be critical for species distributions in this forest.

Reviewer 2 Report

The authors measured tree ring width over time from Scots pine trees that had been cut over a time period spanning several decades. They wanted to see if Scots pine growth had showed any significant trend over the past century, and if this trend depended on what elevation the site was at. They hypothesized that tree growth may have increased or decreased over time, depending on what global change agent was most influential. They used a linear mixed-effects modeling approach to determine that tree growth increased at lower elevations but decreased at higher elevations. Tree age was an important factor as well - older trees showed more decrease at higher elevations. They conclude by highlighting the importance of local environmental factors to drive forest responses to global change.

The topic is an important one - how are trees really responding to all these global change factors that are happening? Unfortunately, this study is not equipped to answer that question.

The ‘local environmental factors’ that are mentioned throughout the paper, including in the concluding sentence, were not examined as part of the study. This really limits the ability of the authors to explain divergence of growth rate with elevation. The paper could be much improved by trying to find sources of environmental data and re-doing the statistical analysis to include them. Without these data, the discussion reads as speculative and uninsightful.

The study takes a narrow view of factors affecting Scots pine growth. Management practices are not considered as a covariate, when in fact, changes in management practices over time can be considerable and lead to changes in growth. Similarly, natural disturbance patterns over time can also influence growth of affected and unaffected trees. Finally, the (potential) issue of selected harvesting and therefore a biased sample is not addressed - the authors need to defend their methodology and convince the reader that the measured growth rates are truly representative of the site and unbiased by factors related to tree selection for harvest.

The methods are lacking critical information needed to evaluate the study. The number of observations and/or degrees of freedom for the mixed models are nowhere to be found. These must be reported either in the text or in a table. It isn’t clear at what level the random effect is determined - is it for each individual tree? These models must be described more fully. I recommend consulting Zuur et al. (2009*) for help with implementing and communicating mixed effects models. Also, my copy of the manuscript is missing part of equation 2 - it appears that only the first line of equation 2 made it onto the page.

I suggest that the authors either add new data to their statistical analysis to try and account for local environmental factors, or re-frame the paper to address a different question that does not require examining environmental data.

*Zuur, Alain, et al. Mixed effects models and extensions in ecology with R. Springer Science & Business Media, 2009.

In the attached pdf, I have provided line-by-line edits that I hope will help the authors if they choose to re-write the paper.

Author Response

Response to Reviewer 2 

Major  issues.

a) The main argument has to do with the experimental design and research methodology.  Specifically the reviewer indicates: “The topic is an important one - how are trees really responding to all these global change factors that are happening? Unfortunately, this study is not equipped to answer that question. The ‘local environmental factors’ that are mentioned throughout the paper, including in the concluding sentence, were not examined as part of the study. This really limits the ability of the authors to explain divergence of growth rate with elevation. The paper could be much improved by trying to find sources of environmental data and re-doing the statistical analysis to include them. Without these data, the discussion reads as speculative and uninsightful.  The study takes a narrow view of factors affecting Scots pine growth. Management practices are not considered as a covariate, when in fact, changes in management practices over time can be considerable and lead to changes in growth. Similarly, natural disturbance patterns over time can also influence growth of affected and unaffected trees. Finally, the (potential) issue of selected harvesting and therefore a biased sample is not addressed - the authors need to defend their methodology and convince the reader that the measured growth rates are truly representative of the site and unbiased by factors related to tree selection for harvest. The methods are lacking critical information needed to evaluate the study. I suggest that the authors either add new data to their statistical analysis to try and account for local environmental factors, or re-frame the paper to

address a different question that does not require examining environmental

data.”

RESPONSE: We thank the reviewer for the time committed to evaluate our work. We disagree however with her/view although it might have been our responsibility for not properly escribing our methods and data. Here in the response letter and in the revised ms we have tried to clarify these issues. Specifically we have tone down statements implying causal relationships between environmental factors and tree growth variations, and instead we have focused on the description of temporal growth trends along the elevation gradient.

The reviewer states "authors measured tree ring width over time from Scots pine trees that had been cut over a time period spanning several decades". This study, however, it is not based on tree rings width but on direct measurement of harvested tree growth volume conducted over a century.  We have clarified this in the method and refer to similar studies conducted with this type of data (see for example Madrigal-Gonzalez J & Zavala MA, 2014. Competition and tree age modulated last century pine growth responses to high frequency of dry years in a water limited forest ecosystem. Agricultural and Forest Meteorology 192-193: 18-26). This approximation is rather unique and it is possible due to an exceptional one century old historical forest archive (see for example further dataset details in Marqués L et al 2018. Last-century forest productivity in a managed dry-edge Scots pine population: the two sides of climate warming. Ecological Applications 28: 95-105). Unlike most dendro studies, direct volume measurements (trees were cut into pieces and the volume of each portion was measured independently) were stratified across pine age classes, thus we can model tree growth dynamics for all size classes. Also by targeting volume rather than diameter or basal area we do not have a bias due to height versus diameter (e.g, trees with low basal area growth but a high stem elongation).

With respect to the major methodological criticism we want to clarify that our approximation is not a multivariate statistical model consisting of regression growth (dependent variable) against drivers (independent variable). This type of studies are of course helpful but usually we lack temporal series of environmental data as long as a century neither spatial variability across a whole forest in detailed climatic and soil data. For this reason in this study we use a statistical model that allows us to describe a pattern of tree growth variation with respect to calendar year and altitudinal gradient. Both time and altitude are surrogates of a number of factor inducing variability. The value of this approach is that although it is based on patterns and correlacionts it can give us insight on mechanisms underlying these patterns. We have done so in the discussion but we are aware that these patterns do not imply causation.

The use of altitudinal gradients have a long tradition in ecology and the use of mixed models to infer patterns in temporal trends is also a widely used method to describe these type of patterns (see for example Pretzsch H. et al. Forest stand dynamics in Central Europe has accelerated 435 since 1870. Nat Commun 2014, 1–10).

b) The number of observations and/or degrees of freedom for the mixed models are nowhere to be found. These must be reported either in the text or in a table. It isn’t clear at what level the random effect is determined - is it for each individual tree? These models must be described more fully. I recommend consulting Zuur et al. (2009*) for help with implementing and communicating mixed effects models. Also, my copy of the manuscript is missing part of equation 2 - it appears that only the first line of equation 2 made it onto the page.

RESPONSE: We have redone the analyses and modified the equation as described below.

Round  2

Reviewer 2 Report

This revision is improved over the first version that was submitted. Essential details of the experimental design and methodology were added to the paper that greatly help the reader interpret the study. I appreciate the care taken to re-run the mixed effects model selection process. There are still a few details that are needed, but these changes should be minor. I added edits and comments to a pdf of the revised manuscript and have attached it to this review form - these should help the authors further improve the manuscript.

One of my comments about the level of the random effects in the mixed effects models does not seem to be addressed. In both models, the authors need to indicate at what level the random effect was assigned - was it tree, or unit? Further, Table 1 has some abbreviations that need to be defined and units of measure must be indicated, and for one of the columns ("Density"), I could not find in the Methods how it was calculated.

Author Response

June 22, 2019

Ms. Brenda Zhao

Assistant Editor

Dear Ms Brenda:

Please find attached the third reviewed version of the manuscript (Forests-521187) entitled "Divergent last century tree growth along an altitudinal gradient in a Pinus sylvestris dry-edge population", by Dr. Laura Fernández-Pérez, Dr. Miguel A. Zavala, Dr.  Pedro Villar-Salvador and Dr. Jaime Madrigal-Gonzalez.  

Response to Reviewer 2:

This revision is improved over the first version that was submitted. Essential details of the experimental design and methodology were added to the paper that greatly help the reader interpret the study. I appreciate the care taken to re-run the mixed effects model selection process. There are still a few details that are needed, but these changes should be minor. I added edits and comments to a pdf of the revised manuscript and have attached it to this review form - these should help the authors further improve the manuscript.

RESPONSE: We thank Reviewer 2 for her/his positive comments and in particular, for text editing, which has significantly clarified the manuscript.

The response to main comments is described below:

One of my comments about the level of the random effects in the mixed effects models does not seem to be addressed. In both models, the authors need to indicate at what level the random effect was assigned - was it tree, or unit?

RESPONSE: We agree with the reviewers that this is an important point. We have added in Line 187 (new version) the following paragraph: “We considered forest unit as a random factor under the assumption that growth of trees in the same forest unit can be correlated due to similar management status and soil conditions”.

Further, Table 1 has some abbreviations that need to be defined and units of measure must be indicated, and for one of the columns ("Density"), I could not find in the Methods how it was calculated.

 RESPONSE: The text now is: Table 1. Characteristics of P. sylvestris in the study area. Density values are number of individuals per ha. Values in parenthesis are standard deviation.

Much of this information is not needed for this paper, because effects of Ca and N fertilizing are not examined in this study. You could shorten this to communicate only the point that previous studies have shown growth increases in response to a range of factors, but it is not clear how elevation or tree size would affect these trends.

RESPONSE: We have simplified this paragraph eliminating references to unnecessary mechanisms and simply describing the key phenomena (i.e. fertilization effects and increasing aridity)

For the rest of the comments inserted in the text, we have edited the text directly in the manuscript of the new version according to her/his suggestions. We have not used a "Track Changes" function in Microsoft Word because we have literally accepted the editions proposed by Reviewer 2 on the pdf file.

 Sincerely,

The authors
